# Recent Advances and Challenges in the Early Diagnosis and Treatment of Preterm Labor

**DOI:** 10.3390/bioengineering11020161

**Published:** 2024-02-06

**Authors:** Prashil Gondane, Sakshi Kumbhakarn, Pritiprasanna Maity, Kausik Kapat

**Affiliations:** 1Department of Medical Devices, National Institute of Pharmaceutical Education and Research Kolkata, 168, Maniktala Main Road, Kankurgachi, Kolkata 700054, India; 2Department of Regenerative Medicine and Cell Biology, Medical University of South Carolina, Charleston, SC 29425, USA; maity@musc.edu

**Keywords:** preterm birth (PTB), prelabor rupture of membrane (PROM), biomarkers, lateral flow immunoassay (LFIA) device, microfluidics, multi-omics

## Abstract

Preterm birth (PTB) is the primary cause of neonatal mortality and long-term disabilities. The unknown mechanism behind PTB makes diagnosis difficult, yet early detection is necessary for controlling and averting related consequences. The primary focus of this work is to provide an overview of the known risk factors associated with preterm labor and the conventional and advanced procedures for early detection of PTB, including multi-omics and artificial intelligence/machine learning (AI/ML)- based approaches. It also discusses the principles of detecting various proteomic biomarkers based on lateral flow immunoassay and microfluidic chips, along with the commercially available point-of-care testing (POCT) devices and associated challenges. After briefing the therapeutic and preventive measures of PTB, this review summarizes with an outlook.

## 1. Introduction

Live births occurring before 37 weeks of pregnancy are referred to as PTB, which is a complicated multifactorial syndrome, defined as idiopathic or spontaneous PTB (sPTB), which makes up 70% of the total PTB, and the remaining cases are medically indicated. PTB resulting from cesarean or labor induction [1]. Preeclampsia, placenta previa, fetal growth retardation, etc., directly endanger the health of the mother or the fetus and are classified as indicated PTBs [2]. According to the WHO, PTB can be classified into extremely preterm (<28 weeks), very preterm (28 to <32 weeks), and moderate to late preterm (32 to 37 weeks), depending on gestational age (Figure 1a) [3]. Premature delivery is the primary cause of death for children under five worldwide [4]. Necrotizing enterocolitis, respiratory distress syndrome, periventricular leukomalacia, seizures, intraventricular hemorrhage, cerebral palsy, hypoxic–ischemic encephalopathy, visual and hearing impairments, infections, feeding issues, and many other short- and long-term morbidities are more common in preterm neonates [5]. The global PTB rate increased from 9.8% to 10.6% between 2000 and 2014, with an expected 13.4 million cases (1 in 10 newborns) in 2020. In 2020, India accounted for 3.02 million PTBs, or almost 23% of all PTBs globally, the most significant number of preterm births worldwide, and the fourth highest PTB rate after Bangladesh, Malawi, and Pakistan (Figure 1b) [6]. Preterm labor (PTL) is “regular uterine contractions before 37 weeks of pregnancy that cause cervical change or regular contractions with an initial presentation with cervical dilation of 2 cm or more” [7]. It is characterized by mild abdominal pains, back pain, regular uterine contractions, watery or bloody vaginal discharge, increased volume of discharge, and the rupture of membranes with water leakage. Various risk factors are responsible for PTL, such as infection (endotoxin), which accounts for 25–40% of cases [8], inflammatory mediators (IL1β, TNF-α) [9], vaginal bleeding (hemorrhage) [10], uterine overdistension [11,12], excessive amniotic fluid volume (polyhydramnios) [13], stress [14], immunological complications, and PROM. PROM, or prelabor fetal membrane rupture, can happen before the regular onset of uterine contractions at term (≥37 + 0 weeks of gestation) or preterm (less than 37 + 0 weeks of gestation), which is also known as preterm PROM (PPROM) [15]. Other significant determinants include the mother’s demographics, gestational age, nutritional condition, history of pregnancy, psychological traits, smoking, alcohol intake, and biological and genetic factors, although the exact mechanism underlying PTB remains unclear [16,17,18,19]. PTL can be diagnosed by analyzing cervical parameters, amniotic fluid, and different biomarkers [7].

The earlier reviews mainly concentrated on establishing a relationship between PTB and adult mortality [20], the impact of high temperatures during pregnancy [21], other risk variables [18,22], and inflammation [23,24]. The others focused on diverse topics, including the estimation of PTB at the national, regional, and global levels [5], the conventional prognostic methods for PTB or PROM and comparative analysis [7,25,26], the clinical practice guidelines (CPGs) for PTB management [27], the impact of progesterone on the PTB rate in asymptomatic singleton and twin-pregnant women [28] or placental alpha microglobulin-1 (PAMG-1) in threatened preterm delivery (TPD) [29], and so on. However, no one has systematically reviewed the guiding principles and evolving technologies for the early detection of PTB and associated POCT devices available in the market. Following an overview of PTB and its current global status, the present review addresses the early diagnosis of PTB using traditional and advanced techniques. It then provides an impression of the POCT devices available for PTB detection, such as microfluidic chips and lateral flow immunoassay test kits. After a brief mention of therapeutic and preventive alternatives for PTB, this review ends with a summary and outlook.

## 2. PTB/PTL Risk Prediction

PTL/PTB risks can be predicted early using conventional and advanced procedures. The conventional techniques involve physical and chemical testing. In contrast, the advanced techniques include multi-omics approaches for detecting biomarkers using a lateral flow or microfluidic device and AI/ML-based approaches for PTL/PTB risk prediction.

### 2.1. Physical Testing

Physical testing includes the speculum examination by observing the color and odor of pooled amniotic fluid [30] and a ferning test to detect “fern-like” crystals on the slide [25] to ensure PROM or the rupture of amniotic membranes. An endovaginal or transvaginal ultrasound (TVUS) can examine parameters like cervical length (CL), cervical dilation, and uterocervical angle (UCA) in the female uterus, ovaries, cervix, and vagina, which are not linked with PROM and are still able to predict the risk of PTL/PTB [31]. The cut-off values used for UCA and CL are ≥110.97° and <3.4 cm for PTL prediction, as listed in Table 1 [32].

### 2.2. Chemical Testing Method

The chemical testing utilizes a nitrazine test for detecting the rupture of amniotic membranes by measuring changes in the pH of vaginal fluid from acidic (pH 3.8–4.5) to alkaline due to the mixing of amniotic fluid (pH ≥ 7.0), indicated by a change in nitrazine paper color from yellow to dark blue. However, there can be other reasons for an increase in vaginal fluid pH to alkaline, which may lead to false positive results [33].

### 2.3. Multi-Omic Biomarker Studies

“Omics” refers to genomics, transcriptomics, proteomics, and metabolomics, and is frequently used to investigate disease biomarkers [34]. Multiple omic studies often predict PTL/PTB risks by finding molecules linked with various pathways (Figure 2a), as listed in Table 2 [35].

#### 2.3.1. Genomic Biomarkers

Genomics study uses sequencing or microarray technology to analyze the gene expression level between gestational periods and sample types. Wnt signaling molecules, genes associated with inflammation and infection, such as EBF1, TIMP2, [36,37] COL4A3 [37,38], TNF [39,40,41,42], and the candidate gene for schizophrenia, i.e., ABCA13, have been identified as PTB biomarkers through the multiple target studies, though their mechanisms remain unclear [43,44,45,46]. TNF receptor genes (TNFR1 and TNFR2), TNFRSF6 gene, and gene variants of Toll-like receptors may be associated with an increased risk of PPROM and PTB [38,41,42,47,48,49]. According to Zhang et al., the genes like WNT4, AGTR2, RAP2C, EEFSEC, and AGTR2 are directly linked to gestational age, while the genes EEFSEC, AGTR2, and EBF1 are associated with PTB [43].

#### 2.3.2. Transcriptomic Biomarkers

Transcriptomics provides information on the abundance of multiple mRNA transcripts in a biological sample [50]. Recently, microRNAs (miRNA) and their mature forms (miR) have been associated with PTB [51,52,53]. miRNAs are non-coding RNAs crucial in regulating gene expression [54]. Earlier studies reported different miRNA levels in PTB [52,53,55,56,57]; for instance, miR-142 and five other miRNAs were identified for inducing shorter gestational duration [53,57]. miRNA transcripts related to EBF1 and MIR4266, MIR3612, MIR1251, and MIR601 are associated with spontaneous preterm birth (sPTB) [58], whereas mRNA, corresponding to Toll-like receptor (4TLR4) and interleukin-6 receptor (IL-6R), can be linked with PTB [59,60].

#### 2.3.3. Proteomic Biomarkers

Numerous protein biomarkers for PTB have been identified by extensive proteomics research [61]. Elevated lipocalin-type prostaglandin D2 synthase (L-PGDS) levels in cervicovaginal fluid (CVF) [58] or inflammatory interleukins (ILs) can cause PTB or PPROM by raising prostaglandin levels, which stimulate smooth muscle contraction in the uterus [62,63,64]. Numerous specific and nonspecific protein biomarkers for PTB have been identified in biological fluids, including amniotic fluid, vaginal secretions, urine, cervical mucus, plasma, and saliva [65]. Fetal fibronectin (fFN), placental alpha macroglobulin-1 (PAMG-1), and phosphorylated insulin-like growth factor binding protein-1 (PhIGFBP1) are considered to be the specific PROM biomarkers for the early prediction of PTB. There are also a few nonspecific biomarkers like ferritin, pregnancy-associated plasma protein-A (PAPP-A), urocortin-1, prolactin, matrix metalloproteases (MMPs), C-reactive protein (CRP), corticotrophin-releasing hormone (CRH), ILs, thrombin–antithrombin (TAT) complex, tumor necrosis factor-α (TNF-α), etc., which are used along with the specific biomarkers for the risk prediction of PTB.

fFN, the extracellular matrix (ECM) protein, is highly concentrated between the trophoblast and decidua, acting as a glue for holding the amniotic sac to the endometrium. PAMG-1 is a placental glycoprotein with a glue-like property similar to fFN [66]. fFN and PAMG-1 can be detected in CVF before 22 weeks of pregnancy and beyond 34 weeks until delivery, and its concentration is either undetectable or falls below 50 ng/mL during the intervening period. Therefore, the fFN level in CVF is monitored to identify the risk of sPTB. If there are no indications of bleeding, cervicovaginal lesions, cervix dilatation <3 cm, or tears of the amniotic membrane, swabs should only be taken for an fFN test before any digital vaginal inspection [67]. The presence of fFN and PAMG-1 during 24 to 34 weeks indicates disruption of the amniotic–endometrium interface, indicating a high risk of PROM. Chen et al. identified phIGFBP1 as a highly sensitive biomarker in CVF for diagnosing PROM (choriodecidual disruption); a positive test indicates a 6.9-fold greater risk of PTB [68]. Usually, IGFBP1, or the non-phosphorylated form of PhIGFBP1, appears in amniotic fluid after 13 weeks of pregnancy, and the degree of phosphorylation persistently increases until the end of pregnancy.

The level of intracellular iron storage protein ferritin ≥37.5 ng/mL during 24–37 weeks of gestation can be correlated with infection, inflammation, and limited expansion of maternal plasma volume, which are associated with an elevated risk of PTL [69,70]. PAPP-A is a growth-promoting enzyme that catalyzes the insulin growth factor (IGF) released by insulin-like growth factor binding protein, facilitating endometrial invasion. It appears in circulation after blastocyst implantation. A low serum concentration of PAPP-A after 24 weeks of pregnancy is significantly correlated with placental dysfunction, stillbirth, PTB, intrauterine or fetal growth restriction, and fetal death [71]. The neuropeptide urocortin, produced by amnion and chorion, encourages prostaglandin-mediated delivery through uterine contraction. A high level of urocortin in the blood and amniotic fluid is associated with PTL, indicating that the peptide has a crucial role in the onset of the condition [72]. Prolactin, a lactation-related polypeptide hormone, is released by chorion, decidua, and amnion, reaching the maximum level (7 μg/mL) in amniotic fluid during the second trimester and staying elevated throughout pregnancy. The detection of prolactin in CVF during 24–36 weeks of pregnancy may indicate decidual membrane rupture, which can be used as a PTB biomarker [73,74]. MMPs are zinc-dependent proteolytic enzymes produced by the placenta and fetal membranes that break down collagen I and IX, and are essential components of the fetal membrane, causing cervical dilatation and fetal membrane rupture [75]. CRP, a nonspecific marker of infection and inflammation secreted by hepatocytes, rises in peripheral blood during amnionitis and intrauterine infection. The CRP level also rises in maternal serum during PTL; therefore, it can be a good predictor of PTB [76,77]. CRH, a placenta-derived hypothalamic peptide, is highly expressed in maternal and fetal plasma. It stimulates parturition through interaction with estrogen, adrenal hormones, prostaglandins, and oxytocin. Studys show that an elevated CRH level (>23.7 pg/mL) can be associated with a high risk of PTB [78]. The precise role of inflammatory markers in PTB is unknown. The pro-inflammatory cytokines, especially IL-6, IL-1β, and TNF-α levels in the blood, can directly link with PTB. TNF-α is produced in maternal and fetal tissues, which regulate immune cells and induce apoptotic cell death [79,80]. TNF-α may serve as a biomarker for the early prediction of PTL and PROM [81,82,83]. A TAT complex is produced after thrombin deactivation as result of coagulation in response to uterine bleeding [84]. An elevated level of TAT is associated with high risk of PTB and can predict PTB with a sensitivity, specificity, PPV, and NPV of 50, 91, 80, and 71%, respectively, with a cut-off value of >8 ng/mL [85]. However, insufficient studies and inconsistent findings limit their clinical diagnostic application. Their diagnostic utility is also less due to their association with multiple diseases.

**Table 1 bioengineering-11-00161-t001:** Methods for predicting PTL/PTB risks and their performance metrics.

Sr. No.	Markers	Sample	Period (Weeks)	Detection Limit	Sensitivity (%)	Specificity (%)	PPV (%)	NPV (%)	Ref.
I. Physical method									
1.	Cervical length	NA	22–24	<25 mm	47	89	37	93	[86]
2.	UCA	NA	18–36	≥111°	65.1	43.6	29.8	77.3	[32]
3.	Ferning test	NA	34–37	NA	84.5	78.2	79.5	83.5	[87]
II. Chemical method									
1.	Nitrazine test	Amniotic fluid	28–36	NA	87.3	80.9	82.1	86.4	[87]
III. Biomarker-based method									
Specific biomarkers									
1.	fFN	CVF	23–34	≥50 µg/mL	66.7	87.9	36.4	96.2	[88]
2.	PAMG-1	CVF	24–34	≥4 pg/mL	90.0	93.8	78.3	97.4	[89]
					66.7	98.6	75	97.9	[90]
3.	IGFBP-1	CVF	20–35	≥30 µg/mL	89.5	94.1	94.4	88.9	[91]
					83.3	84.4	41.7	97.4	[92]
					70	74	48	88	[93]
Nonspecific biomarkers									
1.	Ferritin	Serum		≥37.5 ng/mL	78.7	68.7	71.5	76.3	[70]
2.	CRP	Serum	≤20	≥5.27 mg/L	75	86.1	37.5	96.87	[76]
3.	Prolactin	CVF	24–36	>7 ng/mL	78	80	88.64	64.52	[74]
			20–40	9.5 ng/L	87.03	75	75.80	86.53	[73]
			28–36	30 ng/L	95	78	93	84	[94]
4.	Urocortin-1	Amniotic fluid	13–28	≥57.88 pg/mL	81.8	40.0	40	82	[95]
5.	CRH	Serum	24–36	10.45 pg/mL	80	100	100	55.56	[78]
6.	ACTH	Serum	24–36	14.65 pg/mL	80	100	100	55.56	[78]
7.	MMP-8	Amniotic fluid	20 to 36	>30 ng/mL	82.4	78.0	36.0	97.7	[96]

UCA: uterocervical angle; PAMG-1: placental alpha macroglobulin-1; fFN: fetal fibronectin; CVF: cervicovaginal fluid, CRH: corticotrophin-releasing hormone, CRP: C-reactive protein; MMP: matrix metalloprotease, ACTH: adrenocorticotrophic hormone.

#### 2.3.4. Metabolomic Biomarkers

Metabolomics investigates the cellular metabolites associated with specific biological conditions [97]. The metabolite profile of blood, urine, amniotic fluid, CVF, etc., can be obtained by mass spectrometry analysis followed by nuclear magnetic resonance imaging. Studys show that elevated levels of glutamate, prostaglandins, dulcitol, urocanic acid, N-acetyl glutamine, 1-methyladenine, salicylamide, oleic acid, diglyceride, etc. [38,98,99,100,101], and low levels of glutamine, pyruvate, inositol, alanine, pyroglutamic acid, glutamine, galactose, hexose cluster 3 and 5, inositol, urea, glycerophospholipids (phosphatidylcholines, phosphatidylinositol) and sphingolipids (ceramides), etc., were associated with PTB risks.

#### 2.3.5. Multi-Omic Biomarkers

Stelzer et al. conducted a longitudinal multi-omics study involving metabolome, proteome, and immunome to find putative PTB biomarkers (Table 2) [102]. Using a combination of genomics and proteomics, IL-6 polymorphisms and MMP-9 levels were successfully correlated with PTB [103]. Similarly, gene polymorphisms of TLR4 and TNF-α were linked with TLR4 mRNA level using a combination of transcriptomic and genomic studies, which established elevated TLR4 mRNA expression and may serve as a possible biomarker for PTB [59].

**Table 2 bioengineering-11-00161-t002:** Genomic, transcriptomic, proteomic, and metabolomic biomarkers.

Sr. No.	Identified Biomarkers	Phenotype	Ref.
Genomic biomarkers			
1	ABCA13	PTB	[46]
2	microRNAs (miRNA) and miR	PTB	[51,52,53]
3	TIMP2	Inflammation and infection	[36,37]
4	COL4A3	Inflammation and infection	[37,38]
5	TNF	Inflammation and infection	[39,40,41,42]
6	TNF1 and TNF2	PTB	[42,47,48]
7	TNFRSF6	PPROM	[38,41]
8	Toll-like receptor	PPROM	[43]
Transcriptomic biomarkers			
9	miR-21, miR-142, miR-30e, miR-148b, miR-29b, and miR-223	↓ Gestational period	[53]
10	MIR4266, MIR1251, MIR601, and MIR3612	↑ sPTB risk	[57]
11	LINC00870 and LINC00094	↑ PTB risk	[57]
12	TLR4	↑ PTB risk	[58,59]
13	IL-6R		[60]
Proteomic biomarkers			
14	Lipocalin-type prostaglandin D2 synthase	↑ PTB risk	[104]
15	ILs	↑ PTB and PPROM risk	[62,63]
Metabolomics biomarkers			
16	↑ Glutamate, dulcitol, urocanic acid, N-acetyl glutamine, 1-methyladenine, salicylamide, oleic acid, diglyceride	↑ PTB risk	[38,98,101]
	↓ Glutamine, pyruvate, inositol, alanine, pyroglutamic acid, glutamine, galactose, hexose clusters 5 and 3, inositol, urea, phosphatidylcholines, phosphatidylinositol, ceramides	↑ PTB risk	[38,98,101]
Multi-omics studies			
1	Metabolomic (e.g., arabitol, xylitol, etc.), proteomic (e.g., VEGF 121, activin-A, MMPs, etc.), and immunome (e.g., CD56, INF-α, etc.) markers	Combine metabolome, proteome, and immunome	[102]
2	IL-6 polymorphisms and MMP-9	Combine genomics and proteomics	[103]
3	TLR4 and TNF-α genes with TLR4 mRNA level	Combine transcriptomics and genetics	[59]

PTB: preterm birth; sPTB: spontaneous preterm birth; PPROM: preterm prelabor rupture of the membrane; TNF: tissue necrosis factor; TLR-4: Toll-like receptor 4; INF-α: Interferon α; VEGF 121: vascular endothelial growth factor.

### 2.4. AI/ML Methods

The current methods for predicting PTB are mainly based on hypothesis-based identification of the risks under a controlled set-up, which may include age at pregnancy, multiple gestations, smoking, drugs or alcohol consumption, infections, and chronic illnesses (diabetes, hypertension, obesity, anemia, asthma, and thyroid disease) [105]. However, the results are often misleading, as PTB is also seen in first-time mothers or pregnant women without a known risk factor. Risk prediction can be improved by considering a combination of risks instead of one.

AI/ML utilizes the electronic health record (EHR) or pre-defined clinical risk factors for better predictive performance. Deep learning techniques can also be used to analyze high-dimensional EHR, MRI, and ultrasound data; nevertheless, their predictive efficiency is often limited to 59–75%. Most PTB prediction models are either ineffective at managing the sequential high-dimensional EHR data or lack an appropriate interpretation mechanism that allows them to pick the most significant predictors from the listed variables automatically. Medical data mining involving data collection, description (clustering and association analysis), and prediction (classification and regression) can effectively link PTB with the related risk factors (Figure 2b) [106]. Data are collected through patient surveys; maternal and neonatal records; EHRs; or literature surveys [107]. In data discretization, continuous data attributes are transformed into discretized values (i.e., normalization) by removing outliers and missing data. Optimal features are selected using the filter, wrapper, and embedding methods to enhance the performance of classifiers. Thus, a large amount of data are reduced to a smaller size to make them compatible with the ML algorithm. Finally, model fitting is performed using ML algorithms to select the best classifier among the logistic regression, Naive Bayes, C4.5 decision trees, support vector machines, and neural networks for the automated prediction and diagnosis of PTB. Different metrics are used to assess the performance of the classifier, including precision, recall, F1 score, false positive rate (FPR), and false negative rate (FNR), which are apart from true positive rate (TPR), true negative rate (TNR), and correct classification rate (CCR), representing sensitivity, specificity, and accuracy, respectively [108].

A multivariate logistic regression technique [109,110], deep learning models [111], and artificial neural networks [112] were used to explore various risk factors using 23- and 24-week gestation data based on rules for predicting sPTB or PPROM using ML and data mining [113]. Weber et al. used classifiers for sPTB prediction considering demographics, race–ethnicity, and maternal profile [114], and the best cut-off values of cervical length were calculated for PTB prediction in women with a singleton gestation [115]. On the other hand, Włodarczyk et al. employed ML algorithms based on a convolutional neural network to simultaneously segment and classify the cervix on transvaginal ultrasound images (Figure 2c) [116].

**Figure 2 bioengineering-11-00161-f002:**
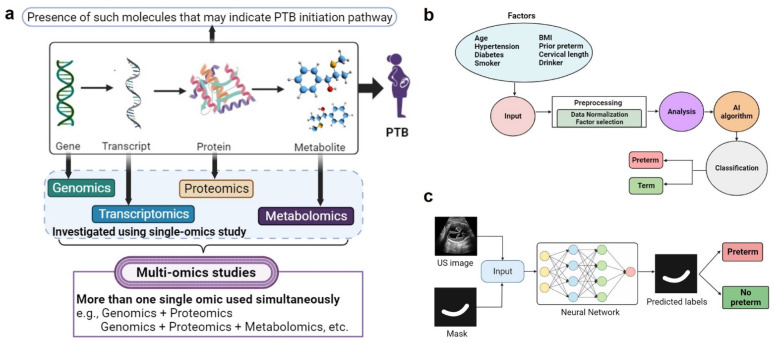
Multi-omics and AI/ML-based approaches for the early detection of PTB. (**a**) The diagram shows that single or multi-omic studies can identify PTB biomarkers containing genes, proteins, and metabolites. (**b**) The workflow shows different stages of AI/ML-based prediction of PTB, data input from electronic health records (EHRs), pre-processing, and classification by a machine learning (ML) algorithm. (**c**) PTB classification using a transvaginal ultrasound image with the mask as an input for the neural network resulting in preterm or the control as an output (inspired from [117]).

## 3. Principles of Biomarker Detection

Lateral flow immunoassay (LFIA) and microfluidic techniques can detect different biomarkers.

### 3.1. Lateral Flow Immunoassay (LFIA)

LFIA, also known as an immunochromatographic or rapid diagnostic test (RDT), is a paper-based analytical technique for the on-site detection of the target analyte in a complex mixture within a few minutes after adding samples to the sample pad [118]. A variety of biological samples can be tested using LFAs, including blood, urine, saliva, sweat, serum, plasma, vaginal secretion, etc. [119]. First, the sample is added to the sample pad on a test strip. Due to capillary flow, the sample migrates to the conjugation pad immobilized with particulate label conjugates, preferably with colored or fluorescent colloidal gold or paramagnetic nanoparticles (Figure 3a). The target analyte is captured as the sample conjugate mixture migrates to the test membrane (porous nitrocellulose), where antigen, protein, antibody, aptamer, etc., are immobilized on test and control lines. Excess samples and reagents continue to flow and get entrapped in the absorbent pad. Positive results are indicated by a colored line on the conjugation pad, which can be interpreted visually or quantitatively by fluorescence-based optical test strip readers [120].

### 3.2. Microfluidic Devices

Microfluidic devices are frequently combined with a “lab-on-a-chip”, made of microscopic channels that transfer fluids from μL to nL in volume. Preconcentration, purification, and labeling are the basic steps for on-chip sample preparation. Polymethyl methacrylate (PMMA) and polydimethylsiloxane (PDMS) are the most widely used materials for fabricating microfluidic devices. Microchip electrophoresis (μCE) is a high-resolution technique that separates ions based on their electrophoretic mobility under an applied voltage [121]. Initially, ferritin, lactoferrin, defensin, TNF-α1, CRF, and other PTB biomarkers were isolated by liquid–liquid extraction and quantitatively estimated by LC/MS, allowing for PTB risk prediction with 87% selectivity and 81% specificity [122]. Alternatively, FITC-labeled preterm birth peptide 1 (P1) and ferritin were successfully analyzed using a miniature pressure-injected multilayer PDMS μCE device, yielding a peak height that was three times greater than the traditional electrokinetic injection [123]. A similar type of pressure-actuated, multichannel, multilayer integrated microfluidic device was created by integrating μCE with upstream SPE, allowing better resolution of peaks for ferritin and corticotropin-releasing factor in the electropherogram [28]. Using a 3D printed microfluidic device with a multiplexed immunoaffinity monolithic column containing specific monoclonal antibodies against each PTB biomarker under fluorescence imaging, three PTB biomarkers—CRF, TNF, and the TAT complex—were successfully eluted from depleted human blood serum (Figure 3b) [124]. The FITC-labelled TAT complex combined with six other PTB risk biomarkers, including PTB peptides 1 and 2 (P1, P2), ferritin, lactoferrin, TNF, and CRF, could be successfully separated and analyzed using a similar “T-shaped” µCE device fitted with a fluorescence detector [125].

**Figure 3 bioengineering-11-00161-f003:**
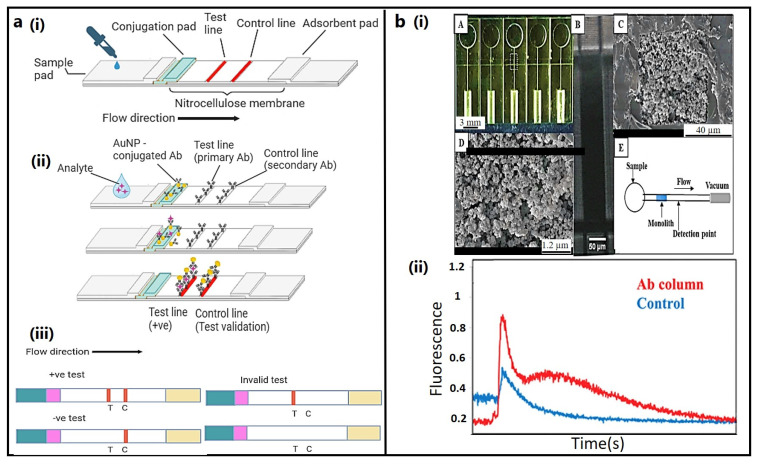
Principles of biomarker detection: (**a**) lateral flow immunoassay (LFIA), (i) test strip design, (ii) representative images of various steps in a sandwich assay, (iii) interpretation of test results; (**b**) 3D printed monolith devices for PTB biomarker extraction (i), 3D printed device (A), monolith within the channel (B), SEM images of the monolith (C, D), and schematics of the device (E), (ii) elution of the labeled biomarkers from the monolith containing antibodies (Ab) against CRF, TNF, and TAT (Ab column) or the monolith lacking Ab (control) producing red and blue fluorescence, respectively [124]. Copyright© 2022 Royal Society of Chemistry.

## 4. Point-of-Care Testing (POCT) Devices

The point-of-care testing (POCT) devices must be “ASSURED” (affordable, sensitive, specific, user friendly, rapid, robust, equipment free, and delivered). The market size for PTB diagnostic test kits is predicted to grow from an estimated USD 419 million in 2022 to USD 857 million by 2032. Hologic, Medixbiochemica, Qiagen N.V, Paraseng Diagnostics, Inc., Hangzhou AllTest Biotech, Biosynex, etc., are a few of the significant players in the PTB diagnostic market for detecting fFN, PAMG-1, and IGFBP-1, IL-6, etc., in CSV or urine samples; a few of them are discussed in detail.

### 4.1. PartoSure^®^ Test

The CE-approved PartoSure^®^ test kit from Paraseng Diagnostics, Inc. (Boston, MA, USA) (Figure 4a) uses goat anti-mouse monoclonal and anti-immunoglobulin antibodies at the test and control regions to detect PAMG-1 and IgG in CVF, respectively, with a limit of detection (LOD) of 1 ng/mL. A few minutes of dipping the PartoSure test strip into an extraction buffer containing CVF from the swab develops the test and control lines. The sensitivity, specificity, and positive and negative predictive values of the PartoSure^®^ kit recorded in different studies are summarized in Table 3 [126]. In a study by Nikolova et al., the efficacy of the PartoSure™ test was determined involving 101 singleton pregnant women (20–36 weeks) with PTL symptoms, intact membrane, and cervical dilatation (≤3 cm) performing the test before the cervical examination following manufacturer’s instructions. By applying the Clopper–Pearson technique with 95% confidence intervals, the PartoSureTM test’s sensitivity, specificity, PPV, and NPV for predicting PTL within seven days were determined to be 90%, 93.8%, 78.3%, and 97.4%, respectively [89].

### 4.2. QuikCheck™ fFN

The Hologic QuikCheck™ fFN is another CE-approved qualitative fFN detection kit (Figure 4b) that uses a mouse monoclonal anti-fFN antibody and an extraction buffer containing gold conjugated goat polyclonal anti-FN antibody to detect fFN in swab containing CVF. A visible test line indicates a positive test, which can be quantified using the Rapid fFN TLi Analyzer (Hologic, Marlborough, MA, USA) within 20 min of development. The remaining unbound complex migrates further and binds with the immobilized plasma fibronectin, forming the control line. The sensitivity, specificity, PPV, NPV, and accuracy of QuikCheck™ fFN were 94.5, 89.1, 89.7, 94.2, and 91.8%, respectively (Table 3) [127].

### 4.3. HealthcheX^®^ Foetal Fibronectin (fFN) Test

The HealthcheX^®^ fFN quick test cassette from Hangzhou AllTest Biotech measures fFN in vaginal secretions against gold-conjugated anti-fFN antibodies immobilized in the test region (Figure 4c). A colored band is formed in the test region if the concentration of fFN is >50 ng/mL (detectable limits), indicating a positive result with a reported sensitivity, specificity, and overall accuracy of 98.1, 98.7, and 98.4%, respectively (Table 3) [128].

### 4.4. Human Fetal Fibronectin XpressCard

Antagen’s Human Fetal Fibronectin XpressCard uses gold-conjugated mouse anti-human fFN antibody immobilized on the T-test line immobilized with an anti-human fFN antibody to detect fFN in fresh morning urine based on the LIFA principle (Figure 4d). A colored band appearing on the test line indicates a positive test result. The reported sensitivity of the device is 10 ng/mL [129].

### 4.5. Actim^®^ Partus

Actim^®^ Partus (Medix Biochemica, Espoo, Finland) is another CE-marked dipstick test device for detecting phIGFBP-1 in cervical secretions for accurate sPTB prediction in symptomatic women. Two monoclonal antibodies against human IGFBP-1 are used in the dipstick: an anti-pIGFBP-1 antibody conjugated with blue latex particles and an anti-pIGFBP-1 antibody immobilized as a test line on the membrane [91]. Blue-colored bands on the test and control lines indicate a positive and valid test [130]. The device’s claimed sensitivity, specificity, NPV, PPV, and diagnostic efficiency are 60, 67.7, 23, 91.3, and 66%, respectively, with an LOD of 10 ng/mL [131].

### 4.6. Premaquick©

Premaquick^®^ (Biosynex, Illkirch-Graffenstaden, France) enables a combined detection of IL-6 and native (non-fragmented) and total (fragmented and native) IGFBP-1. The presence of native IGFBP-1 in CVF signifies cervical ripening and the cervix’s decidual cells lysis without ruptured membranes. In contrast, a fragmented form of IGFBP-1 indicates local proteolytic activity, the leakage of amniotic fluid, and fetal stress due to contraction. IL-6 signifies infection or inflammation in the amniotic and cervicovaginal zones. Premaquick^®^ offers a comprehensive method for detecting myometrium activity, cervical ripening, and inflammation/infection biomarkers for PTL risk assessment. The triple marker combination (IL-6/native phIGFBP-1/total IGFBP-1) provided an accurate prediction of sPTB in threatened singleton pregnancies with a sensitivity, specificity, PPV, NPV, and accuracy of 95.1, 97.5, 97.5, 95.2, and 96.3%, respectively (Table 3) [132].

**Table 3 bioengineering-11-00161-t003:** Performance metrics of POCT devices for analyzing the PTL/PTB proteomic biomarkers.

Sr. No.	Device	Biomarker	Sample	LOD (ng/mL)	Sensitivity (%)	Specificity (%)	PPV (%)	NPV (%)	Accuracy (%)	Ref.
1.	PartoSure^®^ test	PAMG-1	CVS	1.0	80 (<7 d)	95	96	76	-	[126]
					63 (<14 d)	96	89	91	-	
2.	Quikcheck fFN test	fFN	CVS	≥ 50	94.5	89.1	89.7	94.2	91.8	[87]
3.	healthcheX fFN test	fFN	CVS	>50	98.1	98.7	-	-	98.4	[128]
4.	Antagen fFN XpressCard	fFN	Urine	10	-	-	-	-	-	[129]
5.	Actim^®^ Partus	ph IGFBP-1	CVS	10	60	67.7	23	91.3	66	[131]
					95	92	86	97	-	[133]
					80	94	57	98	-	[134]
6.	Premaquick©	IL-6/phIGFBP-1/IGFBP-1	CVS	-	95.1	97.5	97.5	95.2	96.3	[132]

Abbreviations: CVS: cervicovaginal secretion; PAMG-1: placental alpha macroglobulin-1; fFN: fetal fibronectin; ph IGFBP-1: phosphorylated insulin-like growth factor binding protein-1; PPV: positive predictive value; NPV: negative predictive value.

**Figure 4 bioengineering-11-00161-f004:**
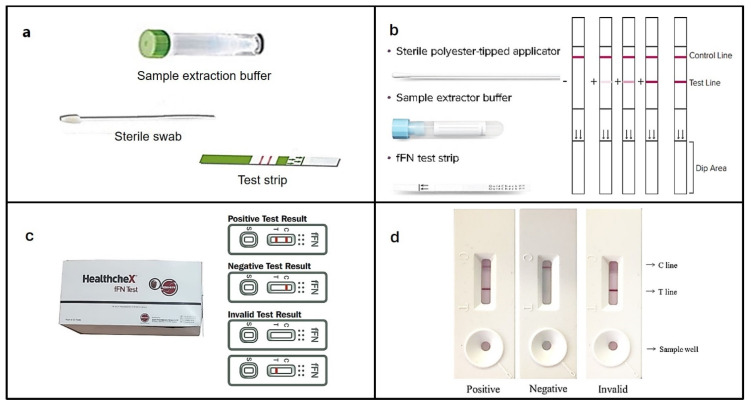
Marketed point-of-care testing (POCT) devices: (**a**) PartoSure^®^ [135] for the detection of PAMG-1, whereas the (**b**) Hologics^®^ QuikCheck fFN test [136], (**c**) HealthcheX^®^ fFN test [128], and (**d**) Antagen^®^ fFN Xpresscard [129] are used for the of detection of fetal fibronectin (fFN).

## 5. Challenges

The early prediction of the PTB is the major challenge that can help to reduce miscarriage cases and the associated PTB complications later in life for premature babies. Half of the women with PTB do not show known clinical risk factors. The current PTB detection methods, such as the TVUS examination, the nitrazine test, etc., fail to detect PTB accurately, which makes PTB a complicated and life-threatening condition for both the mother and fetus. The TSUV often uses a cut-off value of ≤25 mm for CL measurements, but women with a CL of 25–30 mm are still at risk of giving birth within seven days. In that case, combining CL assessment with quantitative fFN testing may improve the predictive capacity [137].

The usage of diagnostic markers has several drawbacks, one of which is the potential for false negative diagnoses, which can increase the fetus’s risk of morbidity and death. The fFN test is frequently interfered with by vaginal bleeding, and it is also inappropriate to screen asymptomatic or multiparous women [138]. Likewise, the cervical phIGFBP-1 test has poor predictive accuracy in asymptomatic patients, and it is frequently impacted by vaginal bleeding, antibiotic use, and sexual activity [139]. While kit-based detection helps predict PTL/PTB risk well in advance, the tests must be run every week beginning at 23 to 34 weeks, which can be expensive as each test costs approximately USD 40, and a pack of ten test kits can cost anywhere from USD 180 to USD 400.

## 6. Treatment and Preventive Measures

“High-risk pregnancy”, a significant contributor to PTL/PTB, is often indicated by a body mass index (BMI) of pregnant women either less than 18.5 (underweight) or 25.0 to 29.9 or even more (overweight or obese), inherited or acquired health issues (diabetes, hypertension), infections, complications from a prior pregnancy, or unexpected causes that may arise during pregnancy [140,141,142]. Perinatal depression, obstetrical risk factors (e.g., multiple gestations, breech presentation, and prior cesarean birth), and pregnancies occurring during teenage years or later in reproductive life are the other risk factors involved in PTL/PTB [143]. Caesareans bear more risk of surgical complications (infection and bleeding), uterine scarring, uterine rupture, and placenta accreta in subsequent pregnancies [144]. Therefore, qualified healthcare providers should thoroughly assess these risks for expectant mothers before recommending a hospital, birth center, or home birth. Though planned home and birth center deliveries are less complicated than hospital births [145], hospitals are the safest place for women to receive intensive care during vaginal birth or an emergency (e.g., cesarean delivery or newborn resuscitation), even though they bear risks of iatrogenic injuries.

Once someone is identified with potential PTB risk, the individual should immediately contact the healthcare provider or, if necessary, be admitted to the hospital. Medications like tocolytics can delay uterine contraction or labor by 24–48 h and avoid preterm parturition [146], and antenatal corticosteroids (betamethasone and dexamethasone) can prevent perinatal complications [147] (e.g., respiratory distress syndrome), or hormonal therapy (e.g., progesterone supplementation) can be used to maintain pregnancy. The PTB rate was significantly reduced by 45% before 33 weeks of gestation for women with short cervical lengths who applied vaginal progesterone gel. This reduction was mainly attributed to the anti-inflammatory properties of progesterone or the elevated local progesterone level in fetal tissues lacking progesterone [148,149].

Magnesium sulfate (MgSO_4_) was routinely used as a tocolytic agent. Still, meta-analyses showed no anti-contraction efficacy, instead exerting harmful effects on the developing fetus if used for more than five days [150,151]. Nevertheless, the WHO and other pediatric and obstetrical societies recommend that using MgSO_4_ was safe for pregnant women at risk of PTB. Additionally, it can also be a cost-effective solution for reducing the risk of cerebral palsy and brain damage, in addition to enhancing the neurological function of preterm babies owing to its antioxidant, anti-inflammatory, and anti-apoptotic activities [152,153]. Other drugs like β-agonists (e.g., terbutaline, Isoxsuprine), calcium channel blockers (e.g., nifedipine), or oxytocin antagonists (e.g., atosiban) can act as tocolytics that relax uterine muscles [154]. NSAIDs (e.g., indomethacin) help suppress the release of prostaglandin, an inflammatory mediator that initiates parturition. Cervical cerclage, which seals the cervical opening with stitches in an emergency during cervical insufficiency at 12–14 weeks of gestation, is one surgical technique that can avoid or delay PTB [155,156,157]. However, it may also prevent the cervix from opening through contractions during delivery.

## 7. Summary and Outlook

The PTB of babies before 37 weeks of pregnancy is one of the leading causes of neonatal mortality and morbidity. This review identifies the conventional and advanced procedures for predicting the PTL/PTB risks early. Among the numerous risk factors, PROM may be the most significant. Speculum examination, a ferning test, and a nitrazine test can mainly detect PROM, whereas TVUS can only detect PTB/PTL risks that may or may not be associated with PROM. Nevertheless, CL evaluation appears to be a more practical method for predicting PTB in standard clinical practice. Genomic biomarkers can indicate both inflammation and infection (e.g., EBF1, TIMP2, COL4A3, TNF) or PROM (e.g., TNFR1 and TNFR2, Toll-like receptors), whereas transcriptomic biomarkers can detect either shorter gestational duration (e.g., miR-142) or sPTB (e.g., EBF1, MIR4266, MIR3612, MIR1251, MIR601, 4TLR4, IL-6R). Similarly, proteomic biomarkers like L-PGDS, ILs, prostaglandin, urocortin, CRH, and TNF-α levels in the blood can indicate uterine contraction leading to PTB. At the same time, fFN, PAMG-1, and PhIGFBP1 are directly associated with PROM. The nonspecific proteomic biomarkers (CRPs) can indicate both, as well as other factors, like placental dysfunction, stillbirth, uterine bleeding, etc. Metabolomic and multi-omic biomarker studies can assess PTB risks with better accuracy.

On the other hand, using AI/ML methods, PTB risk prediction can be statistically made by analyzing the recognized risk factors, such as PROM. Physical tests are often time consuming and produce deceptive results. Multi-omics and quantitative biomarker detection based on an LFIA kit or a microfluidic microchip can be more accurate and sensitive. For PROM detection, fFN resulted in the highest sensitivity, followed by alpha-fetoprotein, CRP, and IL-6. Multiple devices are available for detecting proteomic biomarkers in CSV; however, a combination of biomarkers can improve the prediction accuracy, as observed in Premaquick©. Nevertheless, developing suitable strategies to prevent interferences, enhancing low prediction accuracy in asymptomatic women, and removing false negative results are challenges associated with biomarker-based detection.

The individual with an identified PTB risk should be admitted to a hospital, if necessary, and treated with tocolytics for delaying PTL, corticosteroids for avoiding perinatal complications, or hormonal supplements for maintaining pregnancy. In an emergency, cervical cerclage may be helpful to avoid PTB. Although MgSO_4_ has no known evidence of anti-contraction or smooth muscle relaxant properties, it can be safely used to enhance the neurological function of preterm babies.

## Figures and Tables

**Figure 1 bioengineering-11-00161-f001:**
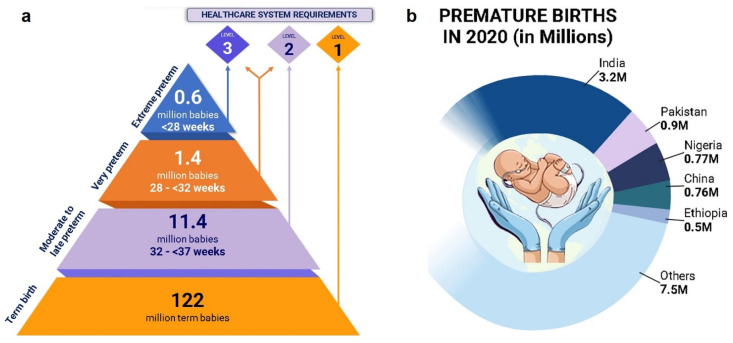
In 2020, there were 13.4 million cases of preterm births (PTBs) around the globe: (**a**) moderate to late, very, and extreme preterm babies requiring essential newborn care (level 1), special newborn care (level 2), and intensive care (level 3), respectively. (**b**) India alone accounted for 3.02 million, the most significant number worldwide (data adopted from [6]).

## Data Availability

Not applicable.

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
