# Peer review of "Recent Advances and Challenges in the Early Diagnosis and Treatment of Preterm Labor"

_bioengineering, 2024, doi:10.3390/bioengineering11020161_

Round 1

Reviewer 1 Report

Comments and Suggestions for Authors

Review paper of the manuscript “Recent advances and challenges in the early diagnosis and treatment of preterm labor”

  1. It´s not necessary to define preterm birth in the abstract.

  2. Pag 1 line 29 - The categories of PTB are not correctly defined. Categories of PTB include late (34 to <37 weeks), moderate (32 to <34 weeks), very preterm (28 to <32 weeks) and extremely preterm (<28 weeks).

  3. Pag 1 line 40  - Preterm labor is not correctly defined. See definition in the cited manuscript (ref 6): “Preterm labor is regular uterine contractions before 37 weeks of pregnancy that cause cervical change or regular contractions with an initial presentation with cervical dilation of 2 cm or more.”

  4. pag 2 line 46 - spell out PROM - premature rupture of membranes (already premature - therefore the abbreviation PPROM is not correct). Figure 1 pag. 2 - What does LAKHS (in the title) means?

  5. Item 2 - pag 2 (line 72) The authors are describing methods to diagnose PROM, but not always PROM leads to preterm labor. It is necessary to separate both and explain that.

  6. pag 4 line 165. Spell out CRP and other abbreviations in the manuscript (ex CRH, TAT, TNF)

  7. Table 1 pag 5. Performance of metrics in predicting preterm labor??? Or in detecting PROM? It is not clear and they are completely different information.

  8. Pag 7 line 206 Spell out AI/ML.

  9. The discussion of the utility of the reviewed methods should be in-deeph

Author Response

1. It´s not necessary to define preterm birth in the abstract.

Answer: The definition of PTB is removed from the abstract.

2. Pag 1 line 29 - The categories of PTB are not correctly defined. Categories of PTB include late (34 to <37 weeks), moderate (32 to <34 weeks), very preterm (28 to <32 weeks) and extremely preterm (<28 weeks).

Answer: According to WHO, PTB can be classified into extremely preterm (<28 weeks), very preterm (28 to <32 weeks), and moderate to late preterm (32 to 37 weeks), depending on gestational age.[2]

3. Pag 1 line 40 - Preterm labor is not correctly defined. See definition in the cited manuscript (ref 6): “Preterm labor is regular uterine contractions before 37 weeks of pregnancy that cause cervical change or regular contractions with an initial presentation with cervical dilation of 2 cm or more.”

Answer: The same definition is introduced in the respective section.

4. pag 2 line 46 - spell out PROM - premature rupture of membranes (already premature - therefore the abbreviation PPROM is not correct). Figure 1 pag. 2 - What does LAKHS (in the title) means?

Answer: Prelabor rupture of the membranes is already abbreviated as PROM in the keywords section.

PROM and PPROM are different. Prelabor rupture of the membranes (PROM) refers to rupture of the fetal membranes prior to the onset of regular uterine contractions. It may occur at term (≥37+0 weeks of gestation) or preterm (<37+0 weeks of gestation); the latter is designated preterm PROM (PPROM). [Scorza, W.E., 2021. Prelabor rupture of membranes at term: Management.]

In Figure 1a (now its Fig. 1b), the population unit is converted from lakhs to million.

5. Item 2 - pag 2 (line 72) The authors are describing methods to diagnose PROM, but not always PROM leads to preterm labor. It is necessary to separate both and explain that.

Answer: The conventional and advanced procedures for early predicting the PTL/PTB risks have been reviewed. Among the numerous risk factors (as mentioned in line 43-50), PROM may be the most significant one. The differences in various detection methods are summarized in Section 7.

6. pag 4 line 165. Spell out CRP and other abbreviations in the manuscript (ex CRH, TAT, TNF)

Answer: C-reactive protein, thrombin-antithrombin complex and tumor necrosis factor-α are already abbreviated as CRP, TAT and TNF-α, respectively in line 129, 130 and 130.

7. Table 1 pag 5. Performance of metrics in predicting preterm labor??? Or in detecting PROM? It is not clear and they are completely different information.

Answer: Actually, the table was incorporated to enlist methods for predicting risks of PTL/PTB, where PROM is one of them, as clarified in point no. 5 and Section 7.

8. Pag 7 line 206 Spell out AI/ML.

Answer: Artificial intelligence/machine learning is already abbreviated as AI/ML in the abstract, in line 17.

9. The discussion of the utility of the reviewed methods should be in-depth.

Answer: The utility of the reviewed methods is discussed in-depth in Section 7.

Reviewer 2 Report

Comments and Suggestions for Authors

An extremely interesting paper. This could be ground breaking in the prevention and management of pre term births. 

However, as a paper, it is too detailed - some unneccessary information on research methodology can be edited. 

Also, there is no mention of any study that has been done to show the sensitivity and specificity etc.  What about challenges - costs, when is the test done, what if someone is identified as PTB potential??

Author Response

1. The paper is too detailed - some unnecessary information on research methodology can be edited.

Answer: As per the reviewer’s suggestions, the unnecessary information from Section 2, 3 and 4 have been deleted to significant extent.

2. There is no mention of any study that has been done to show the sensitivity and specificity etc.

Answer: A study has been included in Section 4.1 to show sensitivity, specificity, PPV, and NPV for predicting PTL. [87] The results of similar studies are listed in Tables 1 and 3.

3. What about challenges - costs, when is the test done, what if someone is identified as PTB potential??

Answer: While kit-based detection helps predict PTL/PTB risk well in advance, the tests must be run every week beginning at 23 to 34 weeks, which can be expensive as each test costs approximately $40, or a pack of ten test kits can cost anywhere from $180 to $400. (as incorporated in section 5. challenges)  

Once someone is identified with potential PTB risk, the individual should immediately contact the healthcare provider or if necessary, they should be admitted to the hospital. Medications like tocolytics can delay uterine contraction and avoid preterm parturition, antenatal corticosteroids can help the fetal lung mature or a hormonal therapy (e.g., progesterone supplementation) can be used to maintain pregnancy. On the other hand, PTL can be avoided by surgical methods such as cervical cerclage, which can stop cervical dilatation. (incorporated in section 6)  

Reviewer 3 Report

Comments and Suggestions for Authors

This article is crowded by information from studies but it is difficult to find authors´ ideas or authors´ aproach to patient in risk of PTL. There are some very argueble information:  magnesium sulphate is used for many years, but the last information show no effect for tocolysis and smart effect as neuroprotective drug pro PLT babies  - cercalge could help in case of cervical isnuficiency, but it is not solution for opening of cervics by contractions. Tables are very nice, but disk graphs in fibure 1b are nonsense. This type of graph are nice for ilustation of distribution of 100% cases not for part of it.

Author Response

1. This article is crowded by information from studies but it is difficult to find authors´ ideas or authors´ approach to patient in risk of PTL.

Answer: The authors have rewritten Section 7 to clarify the preferable diagnostic methods and treatment.

2. There are some very arguable information: magnesium sulphate is used for many years, but the last information show no effect for tocolysis and smart effect as neuroprotective drug pro PLT babies - cercalge could help in case of cervical insufficiency, but it is not solution for opening of cervics by contractions.

Answer: As suggested, section 6 is enriched with the provided information.

3. Tables are very nice, but disk graphs in figure 1b are nonsense. This type of graph are nice for illustration of distribution of 100% cases not for part of it.

Answer: Figure 1b is deleted, and a new figure is introduced in that place.

Round 2

Reviewer 2 Report

Comments and Suggestions for Authors

THank you for editing the paper and explaining some points. Reads better .

I still find it a bit too verbose in certain parts - but I guess if it is for students and researchers to learn from. then it will help. 

It would still be useful to talk a little more about the challenges in practical terms - in the hospital or at home etc. 

Author Response

The authors thank the reviewer for his valuable comments on the manuscript. 

1. I still find it a bit too verbose in certain parts - but I guess if it is for students and researchers to learn from. Then it will help.

Answer: The authors fully agree with this opinion. When searching the existing literature on AI/ML, microfluidic devices, and lateral flow immunoassay, the authors realized they were too technical and difficult for the average reader to grasp the basic concept. Therefore, the authors elucidated the methodologies in detail but in simplified form to provide readers with a comprehensive understanding of the processes by reading this single manuscript.

2. It would still be useful to talk a little more about the challenges in practical terms - in the hospital or at home etc.

Answer: Following the suggestion, we have included more discussions on high-risk pregnancy and challenges in practical terms in home, birth center, and hospital set-up, in Section 6. Treatment and preventive measures (manuscript is highlighted in red and attached in supplementary).

Reviewer 3 Report

Comments and Suggestions for Authors

After revision, it is better and able to be publish.

Author Response

The authors are highly thankful to the reviewer for his insightful comments that helped to enhance the manuscript's quality.